# A Systematic Review of Recreational Nitrous Oxide Use: Implications for Policy, Service Delivery and Individuals

**DOI:** 10.3390/ijerph191811567

**Published:** 2022-09-14

**Authors:** Julaine Allan, Jacqui Cameron, Juliana Bruno

**Affiliations:** 1School of Health and Society, University of Wollongong, Wollongong, NSW 2522, Australia; 2Rural Health Research Institute, Charles Sturt University, Orange, NSW 2800, Australia; 3Department of Social Work, The University of Melbourne, Parkville, VIC 3010, Australia

**Keywords:** nitrous oxide, N_2_O, nangs, systematic review, harm reduction, drug use, substance addiction, health risks, psychological harm

## Abstract

Background: Nitrous oxide (N_2_O) is a dissociative anaesthetic that is sometimes used recreationally. The prevalence of N_2_O use is difficult to quantify but appears to be increasing. Research on N_2_O harms and application of harm reduction strategies are limited. The aim of this mixed method systematic review was to collate and synthesise the disparate body of research on recreational nitrous oxide use to inform harm reduction approaches tailored for young people. Methods: To identify publications reporting the recreational use of N_2_O, a search of public health, psychology and social science databases was conducted. Databases included *PubMed, CIHNAL, PsycINFO, Scopus and Web of Science*. Grey literature and Google advanced search were also used. Due to limited published literature on the recreational use of N_2_O, no limit was placed on publication date or study type. A thematic synthesis extracted descriptive and analytical themes from the selected studies. Quality appraisal was conducted using the CASP Tool for Qualitative studies and the Joanna Briggs Institute case report assessment tool. Results: The search retrieved 407 reports. Thirty-four were included in the final analysis, including sixteen case reports. The included studies were primarily concerned with raising awareness of the apparently increasing use and subsequently increasing harms of recreational N_2_O use. There was limited reference to policy or legislative responses in any published studies, no suggestions for harm reduction strategies or application of service level responses. In general, individuals lack awareness of N_2_O-related harms. Conclusion: The review found three key areas that deserve further consideration including: (1) policy, (2) service delivery, and (3) harm associated with N_2_O use. We recommend a top–down (policy) and bottom–up (services delivery/services users) approach to harm reduction for N_2_O use which also includes further consultation and research with both groups. Future research could explore young people’s experience of N_2_O use including benefits and problems to inform contextually relevant harm reduction strategies.

## 1. Introduction

The pleasurable and intoxicating effects of psychoactive substances result in widespread use. However, deaths and injury from psychoactive substance use, particularly among young people, are a global public health problem [1,2]. Harm reduction is an evidence-based approach to reducing adverse consequences from the use of psychoactive substances, recognising eliminating use is not necessarily realistic or desirable [3]. Harm reduction strategies can be applied in different ways. For example, at a policy level such as the funding of supervised drug consumption sites, at a service level such as gender-specific services or information in different languages; and at an individual level with personalised advice according to the drug used and patterns of use. Drug use occurs on a spectrum ranging from occasional experimentation through to using multiple times a day when highly dependent on a substance. Understanding the benefits and problems associated with different drugs and patterns of use is an important part of creating contextually and physiologically relevant harm reduction strategies [4]. Psychoactive substances vary greatly in the way they change mental states and in potential harms including those caused while intoxicated and others because of regular or prolonged use.

Nitrous oxide (N_2_O) is a dissociative anaesthetic that is sometimes used recreationally. Once inhaled, N_2_O is quickly absorbed into the bloodstream through the lungs; then, it travels rapidly to the brain and the rest of the body. The gas produces a rapid rush of euphoria, heightened consciousness, disassociation, and feelings of floating and excitement, lasting approximately one minute [5]. N_2_O is used within the medical and dental fields for anaesthetic, sedation, and pain relief purposes. Beyond its medical use, N_2_O is used as a fuel booster in the racing industry and in the food industry as an aerosol spray propellant [6].

N_2_O is readily available at supermarkets and convenience stores in small metal bulbs used to produce whipped cream. It is cheap to purchase—each bulb costs approximately $1 AUD. Legalities around the sale of N_2_O vary between countries. For example, there are no legal measures to control the sale of N_2_O in Amsterdam because it is considered relatively safe and moderately used [7]. In contrast, Australia’s states and territories have made it an offence to supply canisters that hold N_2_O to anyone suspected of using it for recreational purposes. However, it is not an offence to inhale it [8]. Australian legal systems have difficulty monitoring the selling or purchasing of N_2_O for recreational use [9]. In the UK, plans to criminalise the use of N_2_O are being discussed because of concerns that use and related harms are growing [10].

The prevalence of N_2_O use is difficult to quantify but appears to be increasing. Misuse among medical and dental professionals because of easy access and methods of controlled administration has been documented [11]. Media reports regarding N_2_O use are common and are fuelling the UK parliament’s concerns about rising prevalence and harms [12]. For example, a ‘Google’ news search on N_2_O brings up 2790 results of news stories related to N_2_O recreational use between 2018 and 2021. The news stories range from high-profile celebrities being caught on camera using N_2_O to young people dying in situations where N_2_O use is implicated (e.g., [13,14]). There are frequent reports in UK local media about parties and events resulting in N_2_O canisters littering public areas.

Most research reporting N_2_O prevalence has used self-selected samples of people who use drugs such as the Global Drug Survey [15] and in Australia, the National Ecstasy and Related Drugs Reporting System [9]. However, both these sources report year on year increases in use of N_2_O and that young people are the most frequent users. It is important to note that N_2_O use can be under-reported due to its short-lived effects, inability to be detected through drug testing and because it is often used with other drugs [16]. However, the observed upward trend is concerning because not only is N_2_O easy to access at a low cost and sold in bulk but because of a lack of information for people using the drug [17].

Research on N_2_O harms and patterns of use are limited. Some studies have identified both minor and chronic harms associated with N_2_O use including death by asphyxiation, psychiatric sequelae such as psychosis and physical symptoms, such as burns or frostbitten skin and the depletion of B12 levels from prolonged use of N_2_O resulting in neurological damage [18,19,20,21]. Other harms are related to intoxication: for example, falling over or losing consciousness whilst under the influence of N_2_O [18]. However, patterns, benefits and social factors related to N_2_O use have not been explored in detail. The lack of information readily available to decision-makers, practitioners, educators, and young people makes it difficult to provide contextualised harm reduction strategies to minimise any risks involved in N_2_O use.

The aim of this mixed method systematic review was to collate and synthesise the disparate body of research on recreational nitrous oxide use to inform harm reduction approaches tailored for young people. Mixed methods systematic reviews combine quantitative data such as prevalence and incidence statistics as well as interpretive data on experience of phenomenon to create breadth and depth of understanding [22]. The inclusiveness of a mixed methods review is particularly useful when the aim is to inform healthcare policy and practice in an area that has limited research [23].

## 2. Methods

### 2.1. Search Strategy

To identify publications reporting the recreational use of N_2_O, a search of public health, psychology and social science databases was conducted. Databases included *PubMed, CIHNAL, PsycINFO, Scopus and Web of Science*. Due to limited published literature on the recreational use of N_2_O, no limit was placed on publication date or study type. Title and abstract were searched using truncation as well as Boolean and proximity operators. The search strategy was initially broad; then, three key terms were combined, N_2_O terms (nangs, hippy crack), substance use terms (substance abuse*, misus*) and effect terms (harm reduction, health effects). The final database search string was “nitrous oxide” OR “laughing gas” OR bulb* OR whippets OR nangs OR nitro OR “hippy crack” AND misus* OR overuse OR addiction OR recreation* OR “recreational use” OR “substance abuse” OR “gas abuse” OR inhalant OR “inhalant abuse” AND “harm reduction” OR “health effects” OR “side effects”. A further search for grey literature was conducted using the advanced Google search engine. The search strategy included English only, Adobe Acrobat (PDF) and no date publication restriction using the terms “nitrous oxide” OR “nangs” AND misus* OR “recreational use” OR “substance abuse” AND “harm reduction” OR “health effects”. Lastly, a manual scan of the reference list of key articles to check for other studies that may have been missed in the database search was completed.

### 2.2. Inclusion and Exclusion Criteria

Studies were included if they contained information related to the recreational use of N_2_O. Studies could be of any design and published at any time. Studies were excluded if the focus was on the medical use of N_2_O, for example, the administration of N_2_O as an anaesthetic pre-surgery and if they were published in a language other than English.

### 2.3. Qualitative Data Analysis

A thematic synthesis extracted descriptive and analytical themes from the selected studies [24]. Quantitative and qualitative findings were integrated because the aim of all studies were similar [23]. That is, the studies all aimed to explore or investigate the prevalence and/or impacts of N_2_O use intending to raise awareness of N_2_O identification and harms. A deductive approach was used for the data analysis organised around reports of recreational N_2_O use harms and any proposed harm reduction strategies at policy, service delivery and individual levels. The analysis involved three steps: coding the results and the key conclusions of the included studies line by line to identify codes related to N_2_O harms or impacts and interventions and then organising the coded information into three themes to provide explanations and recommendations for (1) policy, (2) service delivery and (3) individual factors related to N_2_O use.

### 2.4. Quality Assessment

The title and abstracts of all articles were initially screened independently by JB and JA against inclusion and exclusion criteria. Any discrepancies were resolved blindly by the third reviewer, JC. Quality appraisal was conducted using the CASP Tool for Qualitative studies [25] and the Joanna Briggs Institute case report assessment tool [26]. The CASP tool was chosen as it has been recently used in similar reviews where there is a component of qualitative data and has high rates of acceptability [27,28] and ease of use for multiple assessors [29]. The CASP tool ten-item checklist included variables of study design, data analysis, ethics, bias, results, implications, and usefulness individually rated as ‘*yes, no or unclear*’. Studies with seven or more ‘*yes*’ ratings were given an overall rating of ‘*strong*’. If a study had three or fewer no or unclear ratings, it was designated moderate overall, and weak studies had four or more no or unclear ratings. A ‘*not applicable*’ rating was assigned for criteria that did not apply to the study design (e.g., blinding in studies using data linkage).

Case reports were appraised using the Joanna Briggs Institute case report assessment tool [26]. Previous reviews recommend that case reports be assessed separately to other studies because they detail events and treatment of single patients with previously unreported features [30]. Data extraction was undertaken using a modified version of Torgerson’s extraction table [31]. The extraction table included both qualitative and quantitative variables including aim, design, data collection and analysis methods, results, and conclusions. Variables were extracted from all the included papers by JB. JA and JC each reviewed sixteen included studies, and all three authors discussed any discrepancies until consensus was reached.

## 3. Results

A search of *PubMed, CINHAL, PsycINFO, Scopus and Web of Science* on 25 May 2021, yielded 382 studies. A further ten grey literature sources were selected from advanced Google search, and fifteen studies were hand-picked from reference lists of included studies. All search results were imported into Covidence software (n = 407). Three hundred and six studies were excluded, primarily because they focused on the medical use of N_2_O as opposed to recreational use. A total of (41) articles were selected for full-text reviews; (34) were included in the final analysis. Seven studies were excluded because manuscripts could not be located (n = 3); they focused on the medical use of N_2_O and not recreational use (n = 2), and one publication was an editorial (n = 1). Finally, one study was excluded because it was not English text (n = 1) (Figure 1).

Original research, literature reviews and case reports were included in the review. Key characteristics and aims of included studies and case reports are outlined in Table 1. Only eight of the thirty-four included studies were primary research with participant samples recruited to investigate drug use. Of these, N_2_O use was secondary to the aim of two studies [32,33]. One study utilised qualitative interviews [34], and three were self-report questionnaires [32,35,36]. One study conducted a secondary analysis of mortality data to identify inhalant-related deaths, including those implicating N_2_O use (Leigh & MacLean, 2019) [37]. The other twenty-five publications were case studies or literature reviews. Six included papers were literature reviews of which three were structured systematic or scoping reviews and three were general reviews describing physical, psychological, and social impacts of N_2_O use reported in research and policy documents. Twelve studies were conducted in Europe, four studies were conducted in North America, and eight studies were conducted in Asia/Australia. Only three studies were published prior to 2008; nearly half of all studies were published in 2019 and 2020 (n = 17). Just over half of the study participants were male, and most were aged under thirty years. However, the two large surveys included respondents aged up to sixty years, and the mortality data study did not specify an age limit. Eighteen studies were case reports of people presenting to the hospital with varying symptoms (gait disturbance, numbness in extremities, psychiatric abnormalities) who were subsequently treated for N_2_O misuse. Case reports came from Korea (n = 3), China (n = 3), Canada (n = 2), USA (n = 2), Australia (n = 1), the United Kingdom (n = 1), and six reports did not specify their location.

All sixteen studies (excluding case reports) were assessed for methodological rigour, reporting and overall quality using the CASP as detailed previously in the methods [25]. Half of studies, (n = 8) were rated as strong, and half were rated as weak or moderate (Table 2). The most common reasons for being rated *strong* were a clear aim, well-described methodology and clarity of findings. For example, the correlation between age and N_2_O use was clearly reported [35,45], and the recruitment strategy including consent was detailed (e.g., 36). Six included studies were literature reviews. However, they varied in quality. Three were systematic reviews with a strong overall rating because of clearly defined aims, search strategies and detailed results [17,47,48]. The other three were broad ranging literature reviews with weak overall ratings. No aim, method or analysis was reported in any of these publications [7,43,61]. The study by Drug Science et al. [43], for example, was unable to meet the criteria for the checklists, e.g., appropriate research design, ethics, methodology and clear result. However, this publication is the only one that links media reports with a heightened interest in N_2_O by government and legislators. Similarly, the study of Dutch Moroccan young people received a weak rating but provides insight to N_2_O use within a hard-to-reach cultural group [34]. Ng’s [53] study of New Zealand university students is one of the few N_2_O studies that draws its sample from a general population group instead of people who use drugs. Therefore, studies with weak ratings were included but their findings used with caution in the thematic synthesis, thus contributing differently to the total body of evidence [63].

## 4. Case Report Results

All sixteen case reports were rated for their overall quality against the JBI assessment tool designed to critically appraise case reports [26]. For each case report, key questions were rated as either yes, no, or unclear with an overall appraisal decision to include or exclude. Case reports with three or more no or unclear ratings were excluded from the thematic analysis (Table 3). Four case reports were excluded because of limited detail about the individual’s characteristics, presentation and treatment provided [44,52,56,60].

Of the 16 case reports, most (n = 12) were included due to their clear description of events and symptoms upon presentation [35,38,40,41,42,49,50,51,58,59]. All reports gave a clear description of the post-intervention condition in terms of symptoms or lack of symptoms. One case report did not clearly describe the treatment procedure, including the means of assessing the patient and providing photographs of diagnostic procedures [40]. Just over half of the studies (n = 10) clearly described the demographic characteristics, including the patient’s age, sex, race, diagnosis, prognosis and previous treatments. Only a few studies (n = 5) clearly outlined the patient’s history [39,46,50,51,55,57,62]. For example, ‘*There was no significant medical history or family history of neurological diagnoses, genetic or metabolic disorders… the patient had been trying to aggressively diet... helping the anxiety...*’ ([51], p. 2). A majority (n = 10) of case reports clearly described the patient’s current clinical condition on presentation, including symptoms, frequency, and severity. For example, ([55], p. 1) details in their case report “*...presented to the emergency department for confusion, hallucinations, weakness, and falls... she inhaled up to 600 vials of “whippets” per day for the past several weeks...*”.


The most common reason (n = 11) for receiving a no rating in a JBI item was not clearly describing the patient’s history, including the patients’ medical, family, and psychosocial history [38,42,44,49,52,56,58,59,60]. Moreover, some reports (n = 6) failed to describe the patient characteristics in depth [38,39,44,49,59,60].

All case reports provided recommendations, primarily on the medical impacts resulting from N_2_O misuse. However, only one case report [39] provided long-term follow-up with details of treatment outcomes. In the other examples, there is no information about symptoms and impact of N_2_O use post-hospital treatment or if treatment effects lasted.

## 5. Qualitative Results

The included studies were primarily concerned with raising awareness of the apparently increasing use and subsequently increasing harms of recreational N_2_O use. N_2_O was identified as a very commonly used drug in the UK and US with one study noting prevalence of use in the population up to 38% [33]. All studies except one [43] noted that there was minimal information available to inform policy, service delivery or individuals using N_2_O. This was the conclusion regardless of when or where the study was published. For example, Ng et al. [53] stated that their results show a previously unrecognised high prevalence of recreational N_2_O use in first-year university students who were unaware of risks; and van den Toren [36] undertook their study because of a rapid increase in recreational nitrous oxide use reported in several countries with limited attention in scientific research. Similarly, the case studies published across twenty years all aimed to highlight increasing numbers of hospital presentations with symptoms relating to frequent N_2_O use that could be difficult to diagnose because of clinician’s lack of knowledge or experience with N_2_O (e.g., [40,46]).

### 5.1. Policy Implications

There was limited inclusion of policy or legislative responses in any published studies. Three studies identified ease of purchase as a factor in increasing prevalence including availability in retail shops [54] and online [50] even after the UK’s psychoactive substances legislation. In the EU, restrictions on N_2_O purchase were imposed because of health concerns rather than a relationship to drug-related crime [45]. One study noted the risks of N_2_O use were underestimated in policy and legislative responses to drug use. For example, there was no mention of volatile substance abuse (VSA) including N_2_O, in the UK Government’s 2017 Drug Strategy, although VSA caused the same number of deaths as MDMA [37]. Only one publication suggested harms were overstated [43]. However, this publication was an opinion piece with a weak rating. Drug Science [43] concluded that media campaigns are driving concern, and the problem to be addressed by policy or legislation is canisters littering party sites rather than the use of N_2_O.

That conclusion about lack of harms was not shared. Several studies highlighted that the prevalence appears to be increasing worldwide (e.g., [32,45,62]) and therefore is an issue for policy makers to consider because a small proportion of heavy users are at risk of acute harms [17,33]. Nabben [34] reported an increase in nitrous oxide-related traffic accidents in the Netherlands, noting that most users believe the drug is safe and has no lasting effects and therefore not understanding the risks of driving during or after N_2_O use.

### 5.2. Service Delivery—Assessment, Treatment and Education

N_2_O use risks are generally underestimated. For example, the majority (91.6%, n = 99) of those who had heard of N_2_O were not aware of any side effects associated with its use and believed the drug was safe [7,34,35]. However, Ehirim et al. [35] was the only study with a strong quality rating. Formal assessment of N_2_O use rarely occurs in treatment services. When identified, reports of N_2_O misuse do not fit the criteria for substance dependence and therefore may be underestimated because only two to three DSM-5 criteria are identified during assessment. However, Criterion 1 (Taking the substance in larger amounts or over a longer period than was intended) is positive in 98% of cases included in the review by [47]. The capacity of services to respond to people who use N_2_O is complicated by the lack of a formal screening tool and N_2_O use presenting with other drug use [48]. Furthermore, one study suggests people who inhale volatile substances including N_2_O rarely access treatment services [37]. Leigh and Maclean [37] suggest mental health services might be first point of contact for vulnerable solvent users rather than drug and alcohol services.

Information for young people or clinicians about N_2_O harms appears to be limited or unavailable. Three studies noted N_2_O is not included in drug education programs [7,53,54], and another high-quality study reported young people believe more information/education is needed to raise awareness of harms [35]. The majority of respondents in that study (60%, n = 86) chose the scale value of 10 (extremely important) as to how important it is to educate young people about the effects of N_2_O [35]. In relation to where to intervene, one study suggested that because N_2_O is most often consumed via balloons at festivals and in clubs, this is the best type of intervention site [33]. Another study recommended using influencers in media campaigns with the aim to decrease the risks of heavy nitrous oxide use and improve treatment access [34]. Because driving is impaired for up to 30 min after exposure to N_2_O, information for N_2_O users about risks of traffic accidents was recommended [7].

### 5.3. Individuals

Recreational nitrous oxide use is popular with young people. In one study, most respondents (77.1%, n = 108) had heard of N_2_O and 27.9% (n = 39) had taken N_2_O in the past 12 months [33]. Three studies [35,53,54] found males were more likely to use N_2_O than females. For example, N_2_O was more popular among males at 39.0% (n = 16) compared to females at 24.7% (n = 23) [35], and in another survey, 15% of men and 9% of women in NZ had used N_2_O [53]. However, no statistically significant association between age and N_2_O use was found [35]. That is, N_2_O is not exclusively a young person’s drug. Leigh and Maclean [37] identified an increasing prevalence of deaths from volatile substances including N_2_O and that the age profile of those dying is older than the typical user profile [37]. One high-quality study investigating substance use disorders found that while most users use infrequently, and their use is not associated with significant harm, N_2_O was ‘overused’ by men, adolescents and young adults [47]. Another study confirmed that frequent N_2_O use is associated with hallucinations and confusion and persistent numbness and accidental injury [33]. The quantity associated with problematic N_2_O use varies. Studies identified 300 canisters per week [50], 40–60 per day [40] and 50–100 bulbs taken per session by heavy users to remain intoxicated [7]. Worryingly, regular users appear to be increasing the amount they use both in days of use per month and amount used at each session [9,32]. However, patterns of N_2_O use have not been thoroughly investigated nor have the benefits that prompt use.

The case reports found N_2_O use was typically overlooked, and that people deny using it [60]. In some cases, serious degeneration and limb paralysis is observed before N_2_O problems are identified [42], suggesting scope for earlier identification and intervention. Common presentations indicating problematic N_2_O use include muscle weakness, stumbling and spinal cord problems [41,49]. One person was in a wheelchair for a year before the problem of N_2_O use was diagnosed [42]. Ng [53] suggested that with an increasing proportion of vegetarianism, many young people have a higher risk of nutritional deficiencies, further increasing their risk of subacute combined degeneration of the spinal cord. Other case studies reported strokes [59] and thrombosis [57]. Neuropsychiatric symptoms were also identified [60]. One case study made a connection to psychosis and schizophrenia, concluding that N_2_O use may be a causative factor in the development of psychotic symptoms [58]. However, this was only one study and a speculative finding.

Addressing N_2_O harms for individuals currently relies on medical intervention particularly B12 injections. There was one report of an individual who injected themselves with B12 on the advice of a friend, which resolved their symptoms [44]. However, one-third or 29% of people with N_2_O problems will not have B12 deficiency [56]. The most significant limitation of the case reports is that the outcome of medical interventions is not reported. The follow-up of individuals in case reports is rare and was identified as difficult when people do not return for check-ups [42]. Therefore, ongoing symptoms of misuse and success of interventions is unknown.

## 6. Discussion

### 6.1. Policy Response

This mixed methods review found that there was some limited information available to inform policy, service delivery and individuals about harms related to N_2_O use, and that there is a need to incorporate N_2_O in harm reduction strategies because of the global increasing prevalence of use and the general perception that N_2_O is not harmful. The case studies demonstrate the harms of frequent use and have consistently reported the same physical and neurological symptoms for many years. However, this information does not appear to be distributed to people who use N_2_O or health professionals who care for them. In relation to policy, the same approach should be taken with N_2_O as with other drugs: reduce supply and treat N_2_O problems as a health issue rather than a criminal one [1].

### 6.2. Service Level Response

To improve the service level response to harmful N_2_O use, clinicians, community and treatment agencies need greater awareness of the prevalence and harms of N_2_O use to more readily identify when it is a problem. The inclusion of N_2_O in routine assessment tools and processes is recommended, and awareness campaigns about the risks of driving when affected by N_2_O should be created. For individuals, more information about risks and harms available in accessible places such as music festivals and entertainment venues would be more effective than information at point of treatment such as medical centres or drug and alcohol services.

### 6.3. Research Response

What is apparent is that research on N_2_O use has not changed in the past 20 years, and findings have not been disseminated to the community or health systems. While research interest is increasing, as demonstrated by nearly half of the studies in this review being published since 2019, there are still relatively few studies, and they rarely include population samples or an in-depth exploration of people’s experiences of N_2_O use. There were no intervention studies identified suggesting psychosocial methods of reducing N_2_O use have not been tested.

### 6.4. Strengths and Limitations

A strength of this review is the diverse range of countries that were represented. Several limitations also need to be acknowledged. First, although the CASP is a highly cited method of quality appraisal, it not a universally accepted tool; this also applies to the JBI assessment tool. Secondly, the included studies were limited in quantitative data, and thus, only qualitative data could be reviewed. Our finding should thus be interpreted with caution. Lastly, our review was limited to English-only studies, although only one study was excluded on this basis.

## 7. Conclusions

A review of the N_2_O literature was warranted to explore the contemporary and disparate evidence on this yet largely unexplored area of research. The review found that there are three key areas that deserve further consideration, including (1) policy, (2) service delivery, and (3) harm associated with use. We recommend a top–down (policy) and bottom–up (services delivery/services users) approach to the implementation of harm reduction strategies that also includes further consultation and research with both groups. Future research could explore young people’s experience of N_2_O use including benefits and problems to inform contextually relevant harm reduction strategies.

## Figures and Tables

**Figure 1 ijerph-19-11567-f001:**
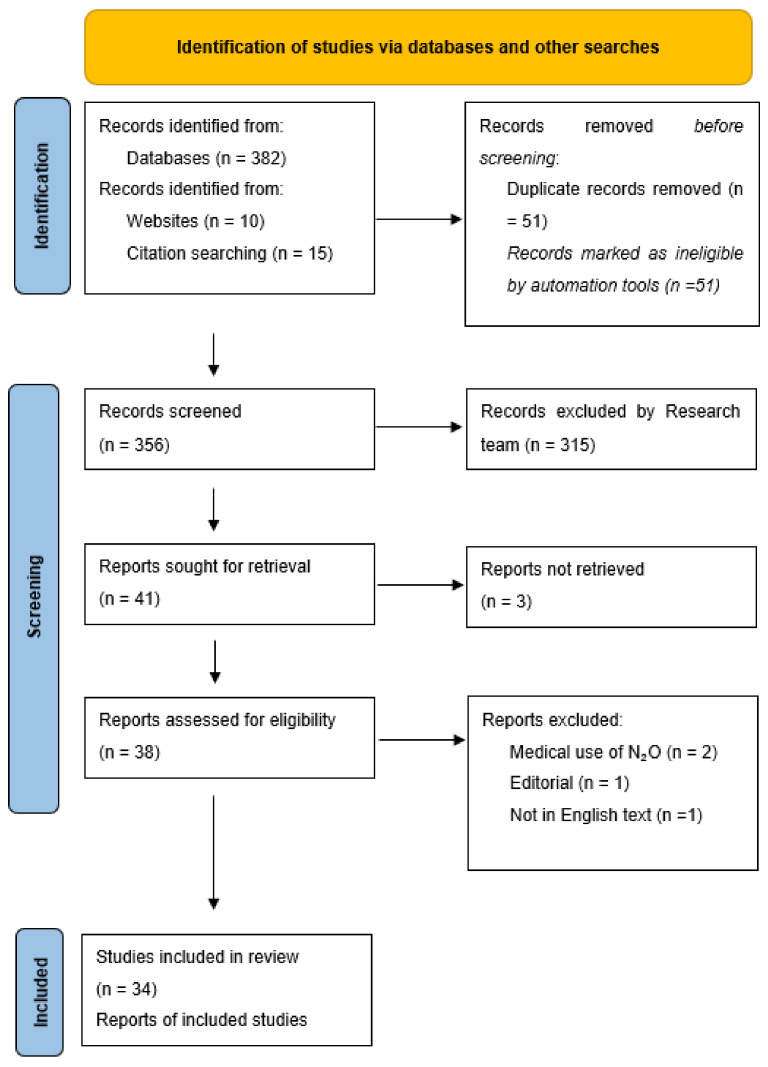
PRISMA Flow Diagram.

**Table 1 ijerph-19-11567-t001:** Study Characteristics.

Authors	Year	Study Design	Country	Study Sample N	Gender (Age)	Study Aim
Asmussen et al. [17]	2020	Scoping review	Denmark and Australia	N = 126 papers	N/A *	To understand current trends and contextualize the public concerns and the new policy responses to N_2_O use
Assaf et al. [38]	2020	Case report	United Kingdom	N = 1	Male (23 years)	To make aware of the potential complications of N_2_O use
Bajaj et al. [39]	2018	Case report	Does not specify	N = 1	Male (23 years)	To report and describe on an ischemic stroke caused from N_2_O abuse
Butzkueven and King [40]	2000	Case report	Australia	N = 1	Male (23 years)	To raise awareness of a potentially serious form of abuse of N_2_O
Chen et al. [41]	2018	Case report	China	N = 1	Male (29 years)	To explore N_2_O case to prompt education and publicity on the harms associated with N_2_O abuse
Choi et al. [42]	2019	Case report	Korea	N = 2	Male (24 years) Female (22 years)	To alert physicians of N_2_O subacute combined degeneration
Drug Science et al. [43]	2020	Literature review	United Kingdom	Not specified	N/A	To inform British MPs of the harms associated N_2_O use
Edigin et al. [44]	2019	Case report	USA	N = 1	Female (27 years)	To raise awareness of possible N_2_O abuse in young patients with neurologic manifestations of B12 deficiency without any other alternative causes
Ehirim et al. [35]	2017	Self-reported surveys	United Kingdom	N = 140	(18–25 years)	To gather information on users and non-users of “hippy crack” among a young population
EUCPN [45]	2021	Questionnaire	Europe	N = 16	N/A	To formulate recommendations that may contribute to the prevention of N_2_O misuse
Fang et al. [46]	2020	Case report	China	N = 66	Males/Females (Mean Age 23 years)	To explore the characteristics of hyperpigmentation in N_2_O users to draw the attention of clinicians to this rare skin condition related to N_2_O abuse
Fidalgo et al. [47]	2019	Systematic review	France	N = 76 papers	N/A	To provide an overview of the substance use disorder potential of N_2_O
Garakani et al. [48]	2016	Systematic review	N/A	N = 77 papers	N/A	To review the case literature and present the neurologic, psychiatric, and medical consequences of N_2_O abuse
Ghobrial et al. [49]	2012	Case report	Not specified	N = 1	Male (19 years)	To make aware the possibility of N_2_O use if cervical trauma is presented for spinal cord injuries
Grigg and Lenton [32]	2020	Survey	Western Australia	N = 797	473 Males/311 Females (Age Range 17 and over)	To provide a greater understanding of the prevalence and correlates of N_2_O use to help inform harm reduction initiatives
Johnson et al. [50]	2018	Case report	Not specified	N = 1	Female (21 years)	To illustrate a rare cause of neurotoxicity resulting from N_2_O misuse
Kaar et al. [33]	2016	Internet survey	United Kingdom USA New Zealand Australia Switzerland Germany	N = 74,804	Not reported (Mean Age 24 years)	To obtain user data to assess N_2_O practice, identify potential risks and minimise harms
Leigh and MacLean [37]	2019	Comparative study	Great Britain	Mortality data 2001–2016	Not reported (Not Reported)	To compare new inhalant (inc. N_2_O) mortality-related data with previous inhalant mortality data
Lundin et al. [51]	2019	Case report	USA	N = 1	Female (22 years)	Bring awareness to the effects of N_2_O
Mo Kin Kwok et al. [52]	2019	Case report	Canada	N = 1	Female (20 years)	To report adverse events to appropriate authorities so that data can be collected about the dangers of N_2_O
Nabben et al. [34]	2021	Exploratory study	The Netherlands	N = 13	10 Males/3 Females (Age 19–28 years)	To explore the beliefs, substance use, health, and youth culture in a Moroccan–Dutch subgroup
Ng et al. [53]	2003	Questionnaire	New Zealand	N = 1782	602 Males/758 Females (Median Age 20 years)	To identify N_2_O awareness and use by university students
Policing and Crime Statistics [54]	2019	Survey	England and Wales	N = 20,500	Adults 16–59 years	To identify the extent and trends in illicit substance use among adults aged 16- to 59-year-olds
Pratt et al. [55]	2020	Case report	Not specified	N = 1	Female (21 years)	To report and describe the authors’ first patient with a venous thrombus associated N_2_O misuse
Seed and Jogia [56]	2020	Case report	Not specified	N = 1	Male (22 years)	To explore N_2_O to support clinicians early identification N_2_O misuse and prompt treatment to support neurological recovery
Sun et al. [57]	2019	Case report	Korea	N = 1	Male (29 years)	To explore N_2_O to support clinician’s early identifying N_2_O and prompt treatment to support neurological recovery
Kim et al. [58]	2019	Case report	Korea	N = 1	Female (22 years)	To discuss the clinical and scientific importance of N_2_O abuse related to acute psychosis
Uil et al. [59]	2018	Case report	Not specified	N = 1	Male (32 years)	To discuss the effect of chronic N_2_O abuse on blood coagulation
Van Amsterdam et al. [7]	2015	Literature review	Not specified	Not specified	N/A	To report actions, effects and impacts related to N_2_O use from published studies
van den Toren et al. [36]	2021	Questionnaire	The Netherlands	N = 555	Male 53%/Female 47%, (Mean Age 15.6)	To identify socio-demographic and health characteristics associated with the frequency of lifetime nitrous oxide use among adolescents
Wong et al. [60]	2014	Case report	Canada	N = 1	Male (30 years)	To consider N_2_O abuse as a cause for neuropsychiatric manifestation
Xiang et al. [61]	2021	Literature review	Not specified	Not specified	N/A	To report effects and prevalence of N_2_O use and its potential complications from published studies
Zheng et al. [62]	2020	Case report	China	N = 43	21 Males/22 Females (Age 15–30)	To improve the awareness of neurological disorders associated with increased prevalence of N_2_O use in China

* N/A—Not Applicable.

**Table 2 ijerph-19-11567-t002:** Study ratings using the CASP criteria.

Study	Clarity of Research Aims ^10^	Appropriate Methodology ^4^	Appropriate Research Design ^9^	Appropriate Recruitment Strategy ^8^	Appropriate Data Collection ^7^	Relationship between Researcher and Participants ^1^	Ethical Considerations ^2^	Rigour of Study ^6^	Clarity of Statement of Findings? ^5^	Value of Research ^3^	Overall Rating
Asmussen Frank et al. (2020) [17]	Yes	Yes	Yes	NA	Yes	NA	NA	Yes	Yes	Yes	Strong
DrugScience et al. (2020) [43]	No	No	Unclear	Unclear	No	unclear	No	No	No	Yes	Weak
Ehirim et al. (2017) [35]	Yes	Yes	Yes	Yes	Yes	Unclear	Yes	Yes	Yes	Unclear	Strong
EUCPN (2021) [45]	Yes	No	unclear	No	unclear	No	No	No	No	Yes	Weak
Fidalgo et al. (2019) [47]	Yes	Yes	Yes	NA	Yes	NA	NA	unclear	Yes	Yes	strong
Garakani et al. (2016) [48]	Yes	Yes	Yes	NA	Yes	NA	NA	unclear	Yes	Yes	strong
Grigg and Lenton (2020) [32]	Yes	Yes	Yes	No	Yes	No	Unclear	Yes	Yes	Yes	Moderate
Kaar et al. (2016) [33]	Yes	Yes	Yes	Yes	Yes	No	Yes	Yes	Yes	Yes	Strong
Leigh and MacLean (2019) [37]	Yes	unclear	Unclear	NA	Yes	NA	NA	unclear	Yes	Yes	Moderate
Nabben et al. (2021) [34]	Yes	Yes	Yes	Yes	Yes	No	No	No	No	Yes	Weak
Ng et al. (2003) [53]	No	Yes	No	Yes	Yes	No	No	No	No	No	Weak
Policing and Crime Statistic (2019) [54]	Yes	Yes	Yes	Yes	Yes	NA	Yes	Yes	Yes	YEs	Strong
Van Amsterdam et al. (2015) [7]	No	No	No	No	No	No	No	No	No	No	Weak
van den Toren et al. (2021) [36]	Yes	Yes	Yes	Yes	Yes	Unclear	Yes	Yes	Yes	Yes	Strong
Xiang et al. (2021) [61]	No	No	No	No	No	No	NO	No	No	No	Weak

^1^ Has the relationship between researcher and participants been adequately considered? ^2^ Have ethical issues been taken into consideration? ^3^ How valuable is the research? ^4^ Is the methodology appropriate? ^5^ Is there a clear statement of findings? ^6^ Was the data analysis sufficiently rigorous? ^7^ Were the data collected in a way that addressed the research issue? ^8^ Was the recruitment strategy appropriate to the aims of the research? ^9^ Was the research design appropriate to address the aims of the research? ^10^ Was there a clear statement of the aims of the research?

**Table 3 ijerph-19-11567-t003:** JBI Quality Assessment of Case Reports.

Study	Patient’s Demographic ^1^	Patient’s History ^2^	Current Clinical Condition ^3^	Diagnostic Tests ^4^	Intervention or Treatment ^5^	Post-Intervention clinical Condition ^6^	Adverse Events ^7^	Takeaway Lessons ^8^	Overall Appraisal ^9^
Assaf (2020) [38]	No	No	Yes	Yes	Yes	Yes	Not applicable	Yes	Include
Bajaj et al. (2018) [39]	No	Yes	Yes	Yes	Yes	Yes	Yes	Yes	Include
Butzkueven and King (2000) [39]	Yes	Unclear	Yes	Yes	No	Yes	Not applicable	Yes	Include
Chen et al. (2018) [41]	Yes	Unclear	Yes	Yes	Yes	Yes	Not applicable	Yes	Include
Choi et al. (2019) [42]	Yes	No	Yes	Unclear	Yes	Yes	Not applicable	Yes	Include
Edigin et al. (2019) [44]	No	No	Yes	Yes	No	Yes	Not applicable	Yes	Exclude
Fang et al. (2020) [46]	Yes	Yes	Yes	Yes	Yes	No	NA	Yes	Include
Ghobrial et al. (2012) [49]	No	No	Yes	Yes	Yes	Yes	Not applicable	Yes	Include
Johnson et al. (2018) [50]	Yes	Yes	Yes	Yes	Yes	Yes	Not applicable	Yes	Include
Lundin et al. (2019) [51]	Yes	Yes	Yes	Yes	Yes	Yes	Yes	Yes	Include
Mo Kin Kwok et al. (2019) [52]	Yes	No	Yes	No	No	Yes	Not applicable	Yes	Exclude
Pratt et al. (2019) [55]	Yes	Yes	Yes	Yes	Yes	Yes	Not applicable	Yes	Include
Seed and Jogia (2020) [56]	Yes	No	No	Yes	No	No	Yes	Yes	Exclude
Sun et al. (2019) [57]	Yes	Yes	Yes	Yes	Yes	Yes	Not applicable	Yes	Include
Kim et al. (2019) [58]	Yes	No	Yes	Yes	Yes	Yes	Not applicable	Yes	Include
Uil et al. (2018) [59]	No	No	Yes	Yes	Yes	Yes	Yes	Yes	Include
Wong et al. (2014) [60]	No	No	Yes	Yes	Yes	Yes	Unclear	Yes	Exclude
Zheng et al. (2020) [62]	Yes	Yes	Yes	Yes	Yes	No	NA	Yes	Include

^1^ Were patient’s demographic characteristics clearly described? ^2^ Was the patient’s history clearly described and presented as a timeline? ^3^ Was the current clinical condition of the patient on presentation clearly described? ^4^ Were diagnostic tests or methods and the results clearly described? ^5^ Was the intervention(s) or treatment procedure(s) clearly described? ^6^ Was the post-intervention clinical condition clearly described? ^7^ Were adverse events (harms) or unanticipated events identified and described? ^8^ Does the case report provide takeaway lessons? ^9^ Overall appraisal.

## Data Availability

Not applicable.

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
