# Peer review of "A Systematic Review of Recreational Nitrous Oxide Use: Implications for Policy, Service Delivery and Individuals"

_ijerph, 2022, doi:10.3390/ijerph191811567_

Round 1

Reviewer 1 Report

Thank you very much for the opportunity to review this article. It is a great pleasure - and I admit a rarity - to read an article of such high methodological quality. The authors not only take up a subject somewhat overlooked in previous studies but, above all, introduce in the detailed, strictly methodological principles of selecting publications for such a review. I admit that I read the manuscript with great admiration for the Authors for the properly selected and solidly described logic of selecting the material for the study and also for its analysis. This is a very well-prepared scientific text and, apart from the need for technical processing, I have no substantive comments.

The clear directions for further research are a valuable indication for policy makers and other researchers.

Author Response

Thank you for your considered comments. We appreciate the support. The manuscript has been edited for grammar and spelling and will be reviewed by the publisher prior to releasing the final version.

Reviewer 2 Report

Firstly, I must commend you on your very interesting and much needed paper. It was very thorough, concise, and of high quality. Overall, there is not really anything crucial that I was left wondering, as all my questions were answered as I proceeded to read. You clearly stated and justified your goals, methodological processes, outcomes, and limits to your research. The structure flowed. Well done!

While not necessarily needed, I was wondering about your stated target for recreational studies in your Introduction. Rather than just defining recreational studies as those outside of medical use, do you have a slightly more detailed definition/statement of what recreational use looks like/includes? For example, in a drug context, do your goals imply regular long-term users as well as short term bingers/social users? I understand that you needed to search recreational use broadly, but the term can have slightly different meanings. As the drug is unfamiliar to many, it may help to provide more specific context on the patterns of N2O use you were investigating.

Kind regards

Author Response

Thank you for your considered comments and supportive feedback. We really appreciate it. We have revised the manuscript according to your suggestions and hope the following addresses your concerns.

Two sentences addressing varied patterns of use have been added to the introduction (lines 44-48 - Drug use occurs on a spectrum ranging from occasional experimentation through to using multiple times a day when highly dependent on a substance. Understanding the benefits and problems associated with different drugs and patterns of use is an important part of creating contextually and physiologically relevant harm reduction strategies (Boyd et al 2020). 

That we do not know about N2O patterns of use has been clarified further to emphasise the aim of the review (line 99 - Research on Nâ‚‚O harms and patterns of use are limited. )

The following sentence has been added to 5.3 Individuals to emphasise the lack of knowledge in this area (line 421 - However, patterns of Nâ‚‚O use have not been thoroughly investigated nor have the benefits that prompt use. )